# Insight on a Competitive Nucleophilic Addition Reaction of Nε-(Carboxymethyl) Lysine or Different Amino Acids with 4-Methylbenzoquinone

**DOI:** 10.3390/foods11101421

**Published:** 2022-05-13

**Authors:** Zhenhui Zhang, Lin Li, Yuting Li, Yi Wu, Xia Zhang, Haiping Qi, Bing Li

**Affiliations:** 1Guangdong Province Key Laboratory for Green Processing of Natural Products and Product Safety, Engineering Research Center of Starch and Plant Protein Deep Processing, Ministry of Education, School of Food Science and Engineering, South China University of Technology, 381 Wushan Road, Guangzhou 510640, China; zhang126zzh@126.com (Z.Z.); lilin@dgut.edu.cn (L.L.); mbb_wuyi@163.com (Y.W.); cexzhang@scut.edu.cn (X.Z.); oliveauspicious@163.com (H.Q.); 2School of Chemical Engineering and Energy Technology, Dongguan University of Technology, College Road 1, Dongguan 523808, China; liyt@dgut.edu.cn

**Keywords:** advanced glycation end products, *ο*-benzoquinone, cyclic voltammetry, coexistence system, binding site, secondary amine, quantum chemistry

## Abstract

*ο*-benzoquinone is a common intermediate which is mostly formed by the oxidation of phenolics or polyphenols containing catechol structure. *ο*-benzoquinone has an outstanding nucleophilic ability, while advanced glycation end products (AGEs) are nucleophilic and can undergo a nucleophilic addition reaction with *ο*-benzoquinone to mitigate the harmful effects of AGEs on the body. As common nucleophilic substances, amino acids existing in large quantities in food processing and in vivo may bind competitively with *ο*-benzoquinone, thus influencing the trapping of *ο*-benzoquinone with AGEs. In this study, cyclic voltammetry and coexistence experiments were used to compare the reactivities of Nε-(carboxymethyl) lysine (CML) and amino acids with 4-methylbenzoquinone (4-MBQ). The results showed that CML is more reactive with *ο*-benzoquinone than most amino acids, and even in complex systems, *ο*-benzoquinone still captured CML. Moreover, almost all adducts were identified by UPLC-QTOF-MS/MS, and their chemical formulas were deduced. Quantum chemistry accurately predicts the efficiency and site of reactions of *ο*-benzoquinone and nucleophiles to a certain extent, and found that a secondary amine has a greater reactivity with 4-MBQ than a primary amine in a similar molecular structure. In general, *ο*-benzoquinone could capture AGEs, thereby showing potential to reduce the harmfulness of AGEs.

## 1. Introduction

Advanced glycation end products (AGEs) are a type of dangerous chemical formed by the Maillard reaction between reducing sugars and amine groups [1]. AGEs can be divided according to their source into dietary AGEs and endogenous AGEs. Although the medical community has not given maximum limits, there is a clear consensus that reducing dietary AGEs intake and endogenous AGEs production can reduce the risk of diseases such as diabetes, cardiovascular disease and kidney disease [2,3,4]. There are many targeted studies on methods of inhibiting the physiological damage caused by AGEs through scavenging free radicals and eliminating dicarbonyls [5,6]. These approaches can reduce the production of AGEs to some extent. However, AGEs are inevitably present in foods or in vivo. To date, reports on eliminating existing AGEs have been rare. Unfortunately, long-term accumulation and cross-linking in vivo contribute to the harmful impact of AGEs [7]. Therefore, it is of great significance to eliminate the generated AGEs.

*ο*-benzoquinone is a common reactive intermediate, mostly formed by oxidation of catechol structure, and has high electrophilic properties [8]. It would easily add to nucleophilic groups on amino acids or AGEs [9], as shown in Figure 1. Epicatechin quinone, which was oxidized by Epicatechin, reacts with the α- or ε-amino group of lysine, thus inhibiting the Maillard reaction [10]. Our previous study found that *ο*-benzoquinone also reacts directly with Nε-(carboxymethyl) lysine (CML), a typical representative of AGEs, forming an adduct of *ο*-benzoquinone and CML [11], which is deduced to be less absorbable in the intestine than free CML. This discovery provides a new potential way to mitigate or eliminate the health hazards of AGEs. The *ο*-benzoquinone structure is ubiquitous in the natural world. For example, catechol, a typical structure found in polyphenols, is easily oxidized into *ο*-benzoquinone, which endows polyphenols with outstanding antioxidant ability. In addition, tyrosinase in the body can convert tyrosine or phenylalanine into dopaquinone, which also has an *ο*-benzoquinone structure and sequentially polymerizes with cysteine or glutathione to form melanin or converts to dopamine [12]. Evidently, regardless of the food system or inside the body, dietary or endogenous AGEs are very likely to bind to compounds with an *ο*-benzoquinone structure, forming AGE-derived adducts that might have different physiochemical properties and biological effects from those of AGEs. However, the binding of *ο*-benzoquinone with AGEs is not specific since the reaction can occur between *ο*-benzoquinone and any other nucleophilic group. Groups with high nucleophilicity would compete with AGEs to bind *ο*-benzoquinone. Among them, amino acids are strong competitors since their amino groups are chemical moieties with strong electron-donating ability and high nucleophilicity. Amino acids are prevalent both in food systems and inside the body, their interference with the binding between *ο*-benzoquinone and AGEs is thus unavoidable. For example, cysteine has been confirmed to have very high reaction rates with *ο*-benzoquinone derivatives [13]. Therefore, to evaluate the binding ability of *ο*-benzoquinone with AGEs and improve the ability of *ο*-benzoquinone to mitigate the health hazards of AGEs, it is necessary to investigate the competitive effect of amino acids on the nucleophilic reaction of *ο*-benzoquinone and AGEs.

To date, there have been few reports on the reactivity of amino acids with *ο*-benzoquinone. Some studies have investigated the reaction kinetics of CML or various amino acids (cysteine, lysine, methionine, glycine, arginine) with *ο*-benzoquinone [11,14,15,16]. However, these studies investigated only the reactivity of individual nucleophilic amino acid with *ο*-benzoquinone, while in food systems or inside the body, AGEs and different amino acids coexist. The binding of *ο*-benzoquinone and AGEs would undergo interference through the combined effects of different amino acids. Thus, it is necessary to analyze the reactivity of each amino acid and AGE with *ο*-benzoquinone in a competitive environment. Moreover, AGEs and some amino acids have more than a single nucleophilic site, so determining the addition site is helpful to clarify the addition mechanism and provide a reference for further application.

In this study, 4-methylcatechol (4-MC), a representative catechol, was employed to react with CML. The reaction of CML and different amino acids with 4-methylbenzoquinone (4-MBQ) formed after the oxidation of 4-MC was studied by cyclic voltammetry (CV). Ultra Performance Liquid Chromatography tandem quadrupole time-of-flight mass spectrometry (UPLC-QTOF-MS/MS) was used to quantify the residual amount of nucleophile to analyze the reactions of nucleophiles coexisting in mixtures with 4-MBQ. The reactivity of nucleophiles with 4-MBQ and the specific sites of the nucleophilic addition reaction were further investigated by mass spectrometry and quantum chemistry.

## 2. Materials and Methods

### 2.1. Materials

4-Methylcatechol (4MC, ≥95%), amino acid mixed standards, and ammonium formate were purchased from Sigma-Aldrich (St. Louis, MO, USA). Twenty amino acids, including L-lysine (Lys, 98%), L-arginine (Arg, 98%), L-alanine (Ala, 98%), L-asparagine (Asn, 98%), L-aspartic acid (Asp, 98%), L-methionine (Met, 99%), glycine (Gly, ≥99%), L-cysteine (Cys, 99%), L-glutamine (Gln, 98%), L-glutamic acid (Glu, 99%), L-histidine (His, 99%), L-isoleucine (Ile, 98%), L-leucine (Leu, 99%), L-phenylalanine (Phe, 98%), L-proline (Pro, 99%), L-serine (Ser, 97%), L-threonine (Thr, 99%), L-tryptophan (Trp, 98%), L-tyrosine (Tyr, 98%), L-valine (Val, 99%), methanol (HPLC grade) and acetonitrile (HPLC grade), were purchased from Aln (Shanghai, China). CML (98%) was purchased from TRC (Toronto, ON, Canada).

### 2.2. Cyclic Voltammetry

A solution of 1 mM 4MC in pH 7.4 PBS buffer (0.2 M) was prepared, and then CML or one of twenty amino acids was added to obtain a specific final concentration. The solubilities of the substances varied, and the concentrations are indicated in the experimental result chart. CV was performed as described in previous studies with some modifications [17]. By fitting the 4MC IpC1 for reduction and nucleophile concentration with an exponential asymptotic curve correlation, the parameters of the fitting formula were recorded, and the integrated area under the curve was determined. The abscissa interval for the integrated area calculation was standardized as 1–64 mM. The area for an amino acid with a concentration less than 64 mM was calculated according to the maximum value of vertical coordinates.

### 2.3. Quantitative Analysis of the Reactivity of 4-MBQ with Nucleophiles by UPLC-MS/MS

A 2.5 mM 4-MBQ solution was produced by the bulk electrolysis of 4MC solution as described in a previous study [13]. The concentration of each amino acid in the mixed amino acid standard was 2.5 mM; these were in 0.1 M HCl solution containing 16 amino acids, not including Cys, Asn, Trp, or Gln. In addition, these four amino acids and CML were blended with the amino acid mixed standard product in 0.1 N HCl to obtain 1.25 mM Cys-containing and Cys-free blended solutions, respectively. The 4-MBQ solution and the combined solution were mixed at molar concentration ratios of 0:1 (control group), 1:1 (low group), 5:1 (medium group), and 20:1 (high group), and the solution was diluted with pH 7.4 PBS buffer to give a nucleophile concentration of 25 μM. The diluted solution was allowed to react at room temperature for 1 h, and then an equal volume of 0.2 M HCL solution was added to stop the reaction.

The reaction product solutions were filtered through a 0.22 μM syringe filter. Samples (2 μL) were injected into a UPLC system (AB Sciex Inc., Framingham, MA, USA) equipped with an Agilent Poroshell 120 HILIC-Z column (2.1 × 100 mm, 2.7 µM) and eluted at a flow rate of 0.5 mL/min with eluent A consisting of 10% ammonium formate in aqueous solution (the pH was adjusted to 3 with formic acid) and eluent B consisting of ammonium formate in acetonitrile solution. Gradient elution was performed as follows: 0 min, 100% B; 7 min, 60% B; 9 min, 60% B; and 10 min, 100% B. The column was operated at 30 °C. Mass spectra were obtained by using an X500R QTOF instrument (AB Sciex Inc., Framingham, MA, USA). The IDA model and MS operating parameters adopted were as follows: curtain gas of 30 psi, nebulizer gas of 45 psi, auxiliary gas of 60 psi, source temperature of 400 °C, scan range of *m*/*z* 100–800, and ion spray voltage of 5500 V for positive mode analytes. Quantification was performed with by the standard external method.

### 2.4. Identification of the 4-MBQ-Nucleophile Reaction Products by UPLC-QTOF-MS/MS

4-MBQ was mixed with CML or amino acids at equimolar concentrations, and then the solution was diluted to 25 μM with pH 7.4 PBS buffer and allowed to react at 25 °C for 1 h. The reaction product solutions were filtered through a 0.22 μM syringe filter. Samples (2 μL) were injected into a UPLC system (AB Sciex Inc., Framingham, MA, USA) equipped with an Agilent InfinityLab Poroshell 120 HILIC-Z column (2.1 × 100 mm, 2.7 µm) and eluted at a flow rate of 0.3 mL/min with eluent A consisting of 0.1% ammonium formic acid and eluent B consisting of methanol. Gradient elution was performed as follows: 0 min, 5% B; 4 min, 5% B; 8 min, 50% B; 10 min, 95% B; 12 min, 95% B; 13 min, 5% B; and 15 min, 5% B. The column was operated at 30 °C. Other mass spectrometry conditions are provided in Section 2.3.

### 2.5. Quantum Chemistry Calculations of the Nucleophilic and Electrophilic Indexes of Reagents

To obtain stable structures, the geometrical parameters of compounds were optimized in the ground state at the DFT/B3LYP level of theory with a 6-311G+(d,p) basis set. The optimized structural parameters were used in vibrational frequency calculations at DFT levels. The final molecular parameters were calculated at the M062X/def2-TZVP level (for single point energy calculations). The calculations were performed with the Gaussian 09 program package version D.01 by invoking gradient geometry optimization using a personal computer. HOMO–LUMO analysis was carried out to explain charge transfer within the molecule. Global hardness (η), global softness (s), electronegativity (χ), and chemical potential (μ) were calculated using the highest occupied molecular orbital (HOMO) and lowest unoccupied molecular orbital (LUMO).

In 1999, Parr and co-authors [18] propose that a specific property of a chemical species, the square of its electronegativity divided by its chemical hardness, be taken as defining its electrophilicity (ω) index (i.e., global electrophilicity index). In 2008, Domingo and co-authors [19] introduce the nucleophilicity (*Ν*) index (i.e., global nucleophilicity index) as:(1)$$N_{\left(Nu\right)}=E_{\mathrm{HOMO}\left(\mathrm{Nu}\right)}\;\left(\mathrm{eV}\right)-E_{\mathrm{HOMO}\left(\mathrm{TCE}\right)}\;\left(\mathrm{eV}\right)$$
where this scale of nucleophilicity is referred to tetracyanoethylene (TCE) taken as a reference since it presents the lowest HOMO energy in a large series of molecules already considered. That is, the N scale is simply the negative of the ionization potential calculated by Koopmans’ theorem with an arbitrary shifting of the origin. This choice allows us conveniently to handle a nucleophilicity scale of positive values.

Site-specific nucleophilicity was compared to determining the Fukui function (FF) [20] and conceptual density functional theory (CDFT) [21]. The Fukui function describes the degree of change in electron density at each position when the number of electrons in the system changes, and it is strictly defined as:(2)$$f\left(r\right)={\left[\frac{\delta \mu }{\delta v\left(r\right)}\right]}_{N}={\left[\frac{\partial \rho \left(r\right)}{\partial N}\right]}_{v\left(r\right)}$$
where *μ* and *N* are the chemical potential and the number of electrons in the current system, respectively, and *ρ(r)* represents the attractive potential of the atomic nucleus for electrons. The Fukui function was condensed to the atom level to obtain the so-called condensed Fukui function (CFF) [22]. Generally, the larger the value of the CFF is at a reaction site, the more likely this site is to be the active site for the corresponding reaction. The CFF describing the nucleophilicity of the specific atom of the molecular is defined as:(3)$$f_{A}^{-}=q_{N-1}^{A}-q_{N}^{A}$$
where *q^A^* is the charge on atom A in the molecule, *q_N_* is the neutral state, and *q_N−1_* is the state lacking one electron in the molecule. When calculating CFF, the Hirshfeld method [23] was used to calculate the atomic charge.

Local electrophilicity index (*ω^A^*) and local nucleophilicity index (*N**_Nu_^A^*) are defined as:(4)$$\omega^{A}=\omega f_{A}^{-}$$
(5)$$N_{Nu}^{A}=N_{Nu}f_{A}^{-}$$

### 2.6. Statistical Analysis

The experimental results were analyzed using SPSS 22.0, and Origin 9.1 was used for correlation fitting. The data were expressed as the mean ± SD from at least three independent experiments. The F-test was used to assess the homogeneity of variance between groups. A value of *p* < 0.05 was considered significant.

## 3. Results

### 3.1. Effects of Nucleophile Concentration on the Reaction of Nucleophiles with ο-Benzoquinone

CV is commonly used to analyze the redox characteristics of substances. Compounds with the catechol structure can form *ο*-benzoquinone during the positive scan and are reduced to catechols during the negative scan [15]. In the presence of nucleophiles, the current intensity of the catechol cathodic peak in the CV experiment is decreased. Therefore, to analyze the reactivity of CML or amino acids with 4-MBQ, the change in the cathode peak of 4MC (EpC_1_ and IpC_1_) after adding different concentrations of nucleophiles, including CML and 20 amino acids, was investigated in the CV experiment, and their reaction dynamic parameters were calculated.

The cyclic voltammograms of 4MC with different concentrations of nucleophiles are shown in Figure 1 and Appendix A. With the addition of nucleophiles, the current intensity of the 4MC cathodic peak (IpC_1_) showed dose-dependent decreases. These results showed that the interaction of the nucleophile with *ο*-benzoquinone which was generated by the oxidation of 4MC occurred, and the reaction efficiency increased with increasing nucleophile concentration. A correlation fitting analysis for reducing the IpC_1_ of 4MC with nucleophile concentration showed an exponential asymptotic correlation that conformed to Equation (6): (6)$$Y=a-b\times c^{x}$$
where the value of *a* can reflect the amount of 4-MBQ reacting in a cycle detection cycle when the amount of nucleophilic reagent tends to be maximum, while the value of *b* represents the fluctuation range of the amount of 4-MBQ reacting in the concentration range of the nucleophiles added. The *c* value is the slope of the fitted curve when the nucleophile is added at the median concentration, and the higher *c* value indicates that the change of nucleophilic reagent concentration will lead to higher change in the amount of 4-MBQ reaction. This fitting results of CML (Figure 1A) was different from Li, et al.’s [17], where a linear correlation was built up. This difference was possibly due to the wider range of CML concentrations selected in our study. The parameter values in Equation (3) are shown in Table 1. Analyzing the reaction efficiency of nucleophiles and 4-MBQ requires a comprehensive evaluation of these three indicators. At the same time, to make the results more intuitive, this study uses the area under the curve (AUC) as a reference indicator. AUC is a common indicator, which is often used to represent the comprehensive characteristics of the fitted curve [24,25,26]. In this study, AUC represents the total reaction efficiency within the concentration range, better reflecting the reactivity of nucleophiles with 4-MBQ.

Comparing the results for reactions with CML and other amino acids showed that CML has a higher a value than most amino acids, indicating that CML has a higher binding efficiency to 4-MBQ. The b value of CML showed that the change in CML concentration had a greater impact on the cathodic peak current of 4MC than that of most nucleophilic amino acids. In addition, the c value of CML was higher than most amino acids, and the AUC value of CML was second only to that of Cys, Pro, or His, far higher than that of its precursor amino acid Lys, which indicates higher reactivity between CML and 4-MBQ. The structural difference between CML and Lys lies in the ε-amino group. The reactivity of CML is higher than that of Lys, probably since the ε-amino group is more active after conversion from a primary amine to a secondary amine [27,28]. Therefore, it appears that the attack on *ο*-benzoquinone came mainly from the ε-amino group of CML, which may eliminate the physiological hazard of CML. In addition, with the addition of CML, Lys, Arg, Pro, Asp, Gln, Phe, Trp, or Leu, a new weak reduction peak EpC_0_ appeared with the reduction peak of 4-MC during the negative scan, which further proved that nucleophiles reacted with the oxidation products of 4-MC to form new complexes.

The reactivity of cysteine is much higher than those of CML and other amino acids. Due to the polarizability and electron-rich nature of sulfur, the sulfhydryl group of Cys is highly nucleophilic [29]. Fortunately, however, Cys is the least abundant amino acid and is often found in the form of cysteine [30]. Its sulfhydryl group is involved in multiple reactions, so it may not be highly competitive with CML in the food system and the body. The reactivity of Pro is second only to that of Cys since Pro has a secondary amine structure similar to that of CML. Brotzel, et al. [27], also confirmed this through photometric experiments. The imidazole group of His also contains a secondary amine, so it is slightly more nucleophilic than CML. While studying detoxification of biomass hydrolysates with nucleophilic amino acids, Xie, et al. [31], found that among twenty amino acids, the ability of His to detoxify biomass hydrolysates was second only to that of Cys, which is thought to be since they both contain secondary nucleophilic functional groups in addition to primary amine groups. According to the results of CV experiments, the reactivities of other amino acids decrease in the order: Val > Ile > Met > Arg > Asn > Lys > Thr > Gly > Phe > Asp > Ala > Trp > Glu > Leu > Ser > Gln > Tyr.

The ability of these amino acids to react with 4-MBQ may be related to the ability of side-chain groups to gain or lose electrons. The general rule is that nucleophiles attack electrophiles and that nucleophiles donate the electrons needed to form a covalent bond [32]. If the side chain of the amino acid is an electron-donating group, it will enhance the electron-donating ability of the α-amino group and improve its nucleophilicity. Studies have confirmed that the alkyl side chains of amino acids have electron donating ability [33,34]. The longer or the more branched it the alkyl chain is, the stronger the electron-donating ability [35], and the higher the nucleophilicity of the α-amino group. Therefore, Val and Ile are significantly more reactive than Ala and Gly. However, Leu shows poor reactivity, although the structure of its side chain is similar to that of Ile, which may be due to different steric hindrances for the side chain groups. The branches of Val and Ile are both at the β-carbon, and the steric hindrance is different from that of Leu. In addition, the inductive effects of alkyl groups are still controversial, and their ability to gain or lose electrons is related to their reaction environment and steric hindrance. The pK_a_ is also an important indicator, and nucleophilic groups with low pK_a_ values release protons easily [36]. For example, the pK_a_ of the amino group of Met is less than that of most amino acids, so its reactivity with *ο*-benzoquinone is higher. The pK_a_ of the Tyr amino group is relatively low, and the electrochemical results showed that Tyr hardly reacts with 4-MBQ. However, solubility may interfere with this result, since only 0.045 g of Tyr dissolves in 100 g of water. The reactions of Asp, Glu, Trp, and Gln are also affected by solubility. Despite their similar structures, Glu and Asn have one more amine reaction site than Gln and Asp, respectively, so their reactivities are slightly higher. Similarly, the high reactivities of Arg, Lys, and Trp also result from this feature. Among them, Trp has the largest and most hydrophobic side chain amino acid [37]; its solubility is only slightly higher than that of Tyr, but it has two amino sites, one of which is a more reactive secondary amine.

### 3.2. The Reactivity of Coexisting Nucleophiles with 4-MBQ

The above results indicate the reactivities of individual amino acids with *ο*-benzoquinone. However, in food and in vivo environments, amino acids coexist with each other. Therefore, it is necessary to analyze the reactivity of each amino acid and CML with *ο*-benzoquinone in a competitive environment. CV can be used to monitor changes in the 4MBQ content of the system in real time, but for a coexistence study, it cannot be used to compare the efficiencies of different nucleophiles in reactions with 4MBQ. Fortunately, the reaction efficiencies of nucleophiles and *ο*-benzoquinone in the coexistence system can be inferred by quantifying the residues of nucleophiles through mass spectrometry. Figure 2A shows that the Cys content was significantly reduced with increasing 4-MBQ addition. At low concentrations, the remaining content of Cys relative to that of the control group was only 21.8%, and at medium and high concentrations, the mass spectrometric signal for Cys could not even be detected; this illustrated the high reactivity of Cys with *ο*-benzoquinone. The residual ratio for CML at low, medium and high concentrations were 97.8%, 87.7%, and 68.8% (*p* < 0.05), respectively. These results further confirmed that CML reacted with *ο*-benzoquinone in a competitive environment. In addition, the reactivity of CML was greater than that of Lys, which proved that conversion of the ε-amino group of Lys into a secondary amine endows CML with higher reactivity. In addition, with increasing addition of 4-MBQ, the content of Arg also decreased significantly, indicating that the reactivity of Arg with 4-MBQ is higher than those of most amino acids. The guanidine group of Arg contains free amine groups, secondary amine groups and alkyl amine groups and forms a conjugated structure, so it exhibits extremely high nucleophilicity. Strangely, this is not consistent with the results of the CV experiments. Our hypothesis is that while detection by cyclic voltammetry is instantaneous, Arg does not fully react with 4-MBQ in one cycle. Other nucleophiles did not show significant decreases in the low concentration group, and only Pro and Gln showed significant decreases at medium concentrations; this indicated that Pro is more nucleophilic than other amino acids and is more likely to react with *ο*-benzoquinone. The secondary amine group may be the key factor in the high reactivity of Pro. However, interestingly, a higher Gln concentration also caused a significant decrease in the reaction efficiency, which may be related to the limited stability of Gln [38]. This is why there was no Gln in the amino acid mixed solution kit. Cys, Asn, and Trp also showed limited stability. In addition, the signal for Gln showed low intensity in the mass spectrum, and the inspection error was significant, which is another important cause of abnormal results. In the high-concentration group, although most amino acids showed a decrease compared with the control group, the contents of His and Ile decreased significantly (*p* < 0.05), and these two amino acids also showed high reactivity in cyclic voltammetry.

Moreover, a comparison between different amino acids at the same concentration showed that there was no significant difference among amino acids in the medium group except for Cys and Gln, which showed that in the system with Cys, low-concentration 4-MBQ reacted preferentially with cysteine, which affected its ability to bind with other nucleophiles. Under high-concentration conditions, the overall trend for the reactions of amino acids was more consistent with the results of the electrochemical experiments. Nevertheless, Tyr showed a higher reactivity than most other amino acids. This was since the solubility of Tyr required testing in the electrochemical experiments with very low concentrations, and its true reaction capacity could not be obtained accurately. Tyr has a low pK_a_ and a phenolic hydroxyl group, and the hydroxyl group of the phenolic ring is strongly electron-donating [39]. Therefore, the amino group of Tyr is strongly electron-donating, which facilitates nucleophilic addition to *ο*-benzoquinone.

The contents of cysteine in food systems and the body are relatively low. Therefore, this study further investigated the ability of each amino acid and CML to react with 4-MBQ in a system without cysteine (shown in Figure 2B). In the low-concentration group, compared with the Cys-containing system, the concentrations of Arg, Pro and CML exhibited more significant reductions, 91.8%, 89.9%, and 82.2%, respectively, than those of other amino acids. In the high concentration group, the residual ratio of the three were reduced to 56.1%, 64.1%, and 51.9%, respectively, consistent with a secondary amine structure with high nucleophilicity. Tyr exhibited good reactivity at medium concentrations, which further showed that its solubility influenced the results of the electrochemical experiments. Under high concentration conditions, compared with the group containing Cys, more amino acids showed significant differences relative to the control group, which implied that the presence of Cys in the system will greatly inhibit binding of o-benzoquinone to CML or other amino acids. In general, the coexistence experiment showed that CML has the ability to bind *ο*-benzoquinone and is superior to most amino acids, regardless of whether the system contains Cys.

### 3.3. Identification of 4-MBQ-Nucleophile Reaction Products by UPLC-QTOF-MS/MS

To further confirm the nucleophilic addition reactions of amino acids or CML with *ο*-benzoquinone, their adducts with *ο*-benzoquinone were detected by mass spectrometry, as shown in Appendix A and in Table 2. The detected product ions for the CML-4-MBQ adduct were at *m*/*z* 206.1187, 152.0717, 130.0874, and 84.0815, and the structural formula was subsequently deduced to prove that the main binding site for 4-MBQ and CML was at the secondary amine of CML. From the product ions, it can be inferred that the binding site of Lys with 4-MBQ should be in the α amino group, which is different from the findings of Li, et al. [16]. The product ions of *m*/*z* 178.9184, 136.9303, and 118.9184 were not detected in their spectra. However, these ions are the key to determining the binding sites, and the quantum chemical data in our study (shown in Figure 3) also prove that the α amino group of lysine has higher nucleophilicity than the ε amino group. Another proof occurred with Trp, which also has two amino sites. MS/MS data for the Trp-4MC adduct showed that a product ion with *m*/*z* 252.0984 was detected, which proved that binding occurred at the secondary amine site of Trp. His also contains a secondary amine structure, and the presence of a fragment ion with *m*/*z* 217.0914 proved that binding occurred with the secondary amine. Moreover, the addition products of L-Ile-4MC and L-Asp-4-MBQ showed a signal only for the precursor ion, and no related product ions were found. This may be related to the instability of the addition product, which would result in low precursor ion abundance and make the product ions difficult to find. For example, the parent ion of a complex formed by 4-MBQ and two Asp was found. The parent ions and product ions of other amino acids were also found, and possible structures and fragmentation methods were deduced, as shown in Appendix A.

### 3.4. Quantum Chemical Prediction of the Reactivity of 4-MBQ with Nucleophiles

Quantum chemistry is also a method for judging the nucleophilicity and electrophilicity of molecules. The nucleophilicity and electrophilicity of molecules can be calculated with the CFF [20,21]. As shown in Table 3, the global nucleophilic index of Cys was slightly lower than that of Met, which was completely different from the results described above. This is due to the fact that the S atom of Met is saturated and has no site available for a reaction with *ο*-benzoquinone. Nevertheless, Met gives a higher calculated result for the global nucleophilic index. The nucleophilic index of Tyr is higher than those of most nucleophiles, which also confirmed that the results of electrochemical experiments were influenced by solubility. In addition, the calculated global nucleophilic index for CML was lower than those of most amino acids, such as lysine, but its global electrophilicity was much higher than those of other amino acids. This does not match the actual efficiency of CML reactions. This may be due to the fact that global nucleophilicity and electrophilicity indicate the overall abilities of molecules to gain or lose electrons and their ability to attract other nucleophilic and electrophilic molecules. Therefore, the high electrophilicity of CML may increase its ability to attract the nucleophilic *ο*-benzoquinone. Thereafter, the strong local nucleophilicity exhibited by the ε-amino group of CML (shown in Figure 3) would improve the binding efficiency of CML and *ο*-benzoquinone. Therefore, in general, quantum chemical computations can predict the nucleophilicity and electrophilicity of molecules to a certain extent. Nevertheless, the actual efficiency for reactions between molecules are also restricted by many factors.

The local nucleophilicity of amino acids containing multiple nucleophilic sites were also calculated with quantum chemistry (shown in Figure 3). These data cannot accurately compare the nucleophilicity and electrophilicity of different substances, but they can compare different sites in the same substance. According to these results, the local nucleophilic index for the S position of Cys is much higher than that for the N2 position, which once again proves that the main reaction site of Cys is its sulfhydryl group. The nucleophilic index of the ε-amino group in CML is much higher than that of the α-amino group, which further proves that the ε-amino group is the main reaction site. The local nucleophilic index of the nitrogen on the imine of Arg is significantly higher than that at the other position, indicating that the imine is the main nucleophilic site, which is consistent with expectations. The N7 position of His and the N7 position of Trp also have secondary amine structures. The local nucleophilic indexes are much higher than that of a primary amine site, but the nucleophilic index of His N9 is also very high. However, His N9 is saturated and cannot undergo nucleophilic addition reactions. Other amino acids with multiple amino groups also have different local nucleophilic indexes, and these dates are in good agreement with the structure predicted by mass spectrometry.

Some typical AGEs have also been studied by using quantum chemistry to predict their reactivity with *ο*-benzoquinone (shown in Table 3 and Figure 3). Due to the similar structures of CML and CEL, their nucleophilic and electrophilic indexes are relatively close. The global nucleophilic index of pentosidine is much higher than that of other AGEs since pentosidine has 6 amino sites, including a secondary amine structure that has the highest local nucleophilic index. Therefore, it is predicted that pentosidine is more likely to react with *ο*-benzoquinone, while CML and CEL show similar reactivities. It should be noted that this conclusion is simply deduced from calculation, which can only be used as a reference, and the actual response ability needs to be further proved. In addition, the local electrophilic index of each site in 4MC was also calculated (shown in Figure 3). The C5 position has a higher electrophilic index, so this site should be the primary reactive site. Therefore, the inferred binding sites of *ο*-benzoquinone in some studies may be inaccurate [10,16,17]. In general, quantum chemistry can predict the efficiency and site of reactions of *ο*-benzoquinone and nucleophiles to a certain extent, which could be helpful for experimentalists. However, it is necessary to comprehensively analyze the global and local site indexes, and the actual reaction efficiency is also restricted by other aspects.

## 4. Conclusions

In conclusion, CV experiments confirmed that CML showed good reactivity compared to most amino acids, and the secondary amine structure greatly enhanced the reactivities of nucleophiles with *ο*-benzoquinone. Coexistence experiments showed that even in complex systems, *ο*-benzoquinone still captured CML. The structures of nucleophile-4-MBQ adducts were predicted from the MS and MS/MS results, and these predicted structures are in good agreement with the results calculated by quantum chemistry. In general, a series of experiments proved that *ο*-benzoquinone could capture CML in single system or complex system, thereby showing potential to reduce the harmfulness of AGEs. Researchers can use this feature to investigate the actual impact of capture behavior on health.

## Data Availability

The data presented in this study are available on request from the corresponding author.

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
