# Peer review of "Insight on a Competitive Nucleophilic Addition Reaction of Nε-(Carboxymethyl) Lysine or Different Amino Acids with 4-Methylbenzoquinone"

_foods, 2022, doi:10.3390/foods11101421_

Round 1

Reviewer 1 Report

The manuscript by Li and co-authors deals with experimental and computational study on a reaction between Nepsilon-(carboxymethyl) lysine (CML) and mixture of various amino acids with 4-methylbenzoquinone (4-MBQ). The main idea of this work is to investigate if CML can be captured by 4-MBQ in the presence of amino acids. CML is a representative of potentially dangerous advanced glycation end products, and its removal is a quite important problem. Motivation of this study is well justified in the introduction. Methods used are generally adequate.  A major revision is suggested in accord with comments below.

  1. General and specific formulae of AGEs, CML and 4-MBQ should appear in introduction. Names of these species may say little to non-specialists in this area. Also it is not clear what reaction was investigated. Is it classical chemical reaction between CML and 4-MBQ/amino acids or this is just formation of adducts due to non-covalent interactions? In the first case, reaction scheme should be presented.
  2. Page 4, line 155. Traditionally, other letters are used to denote hardness, softness, electronegativity and chemical potential (Greek eta, s, Greek chi and Greek mu, respectively).
  3. Page 4, line 167. “… describing the nucleophilicity of a molecule…”. CFF describes the nucleophilicity of a specific center but not a whole molecule.
  4. Figure 1. Point 0,0 should be included in the plots a)-f).
  5. In various places of the manuscript the authors mention the reaction rate assigning it to the coefficient c in Figure 1. I think this is not correct. Reaction rate is a concentration change per time unit and it may be determined from a dependence property-vs-time. No such a dependence is discussed here. In Figure 1, there are dependences of current reduction vs. concentration but not time. The experimental data do not have anything with reaction rates.
  6. Page 7, lines 250-252. “If the side chain group has a strong ability to accept electrons … such as … other amino acids with side chain alkyl groups.” But alkyl groups are not electron acceptors.
  7. Figure 2. What property is along the y axis? It is not clear at all. Certainly, this should not be a rate.
  8. Figure 2 (I). Cyc -> Cys
  9. Figure 2. What do indices A, B, a, b, c, etc. mean here?
  10. Page 11, Table 3. Global nucleophilicity and electrophilicity indices should be defined in Computational details.
  11. Figure 3. What are local nucleophilicity and electrophilicity indices here? Are they Fukui functions? This should be clearly indicated. if fA(+) was used as local electrophilicity index, this should be defined in computational details. Also local softnesses are better local indices than FF. I suggest to use the former.
  12. Conclusions. “The structures of nucleophile-4-MBQ adducts were successfully inferred…” Structures were not inferred at all.
  13. Conclusions. “In general, a series of experiments proved that o-benzoquinone could capture CML in food processing and in vivo…” This study has nothing with in vivo conditions. Certainly, these conclusions cannot be extended to in vivo.

Reviewer 2 Report

The authors studied the reaction of Nε-(carboxymethyl) lysine (CML) and different amino acids with 4-methylbenzoquinone (4-MBQ) both experimentally and theoretically. I find this study well-written and interesting work supported by experimentally measured and calculated results. The paper is clear and easy to follow. I would propose some minor modifications which the authors should take into account during the revision of the manuscript.

The citations are in the sentences and not after: e.g. "Maillard reaction.[1] " (Line 33) -> "Maillard reaction [1]."

Line 45: " has strong electrophilic ability". Check this statement. Please, either verify or change it.

Please, clarify the abbreviation “UPLC-QTOF-MS/MS“ in the last paragraph of the introduction (Line 86).

In the Cyclic voltammetry section, (line 111 ). What does “maximum y value “ refer to?

Please, clarify line 151: "The final molecular parameters were calculated at the M062X/def2-TZVP 151 level."
Do you refer to single point calculations or reoptimization and frequency calculation?

A little bit more about the calculation of local electrophilic indexes would be beneficial for the readers.

In Figure 1, in my opinion putting the images of the concentrations of nucleophiles “ (A: CML; B: Lys; C: Cys; D: Arg; E: Pro; F: Met) “ below each other will be more clear, since you referred to it with the capital letter. The same applies for the “ (a: CML; b: Lys; b: Cys; d: Arg; e: Pro; f: Met) “.

In section “3.2. The reactivity of coexisting nucleophiles with 4-MBQ.“, please, start with text and add the figure at the point of the first reference to it.

I appreciate that you indicated in the description of “Figure 3“ what the box's colors mean. But what I miss that why you did not refer to different colors for the abbreviations of the structures (red, blue, and green). Does it refer to something or it's just the design for the figure? if it does not have any scientific sense, then please make all abbreviations of the structures have the same colors.

The description of the supporting information in the manuscript which starts from “line 439 to line 476“ is too detailed. Please try to make it shortened.

The main conclusion of the research is that the CML is more reactive with o-benzoquinine than the other amino acids. I would like to ask the author to explain this result. Why o-benzoquinine is more reactive than the other amino acids? What is the chemical explanataion behind this statement?  

Round 2

Reviewer 1 Report

In the revised version some of my comments were considered but several important issues were not addressed.

1. I cannot understand how reaction rate was calculated from the plots "amount of reduced substrate vs. concentration". To evaluate reaction rate, time dependences should be analysed.

2. Yes, formulae and general information about the reaction are indicated in Figure 3 and somewhere else. But readers may want to understand the systems and reactions under study immediately, in introduction. I strongly suggest to add this information to introduction.

3. I do not understand meaning of the term "reaction amount" introduced in the revised version. What is it?

4. "the ratio of the content of nucleophilic reagent after adding 4-MBQ to that without adding 4-MBQ" in caption to Figure 2 is not "amino acid resudual rate" as indicated along the y axis. Again, I have very significant doubts about the using term "rate" in this work.

5. Sorry, with all explanations in the caption to Figure 2, I do not understand the meaning of the capital and small letters in this figure.

Author Response

Responses to Reviewer 1 comments (round 2)

We thank the reviewer for the insightful comments on our manuscript. We have addressed all the specific comments and listed the responses below. Changes and corrections in the text are trackable in the revised manuscript submitted to Foods

Point 1: I cannot understand how reaction rate was calculated from the plots "amount of reduced substrate vs. concentration". To evaluate reaction rate, time dependences should be analysed.

Response 1: Thank you for your suggestion. The word "rate" is inaccurate, I'm very sorry. The “c” value here is actually the "slope" of the fitted curve at the median concentration. This error has been corrected in the manuscript.

Point 2: Yes, formulae and general information about the reaction are indicated in Figure 3 and somewhere else. But readers may want to understand the systems and reactions under study immediately, in introduction. I strongly suggest to add this information to introduction.

Response 2: Thank you for your suggestion. I have added Scheme 1 to the manuscript to demonstrate the relevant reaction process.

Point 3: I do not understand meaning of the term "reaction amount" introduced in the revised version. What is it?

Response 3: Sorry, the expression is wrong, it has been corrected in the manuscript.

Point 4: "the ratio of the content of nucleophilic reagent after adding 4-MBQ to that without adding 4-MBQ" in caption to Figure 2 is not "amino acid resudual rate" as indicated along the y axis. Again, I have very significant doubts about the using term "rate" in this work.

Response 4: Thank you for your suggestion. The annotation of the y-axis has been corrected, and the word "rate" has been used incorrectly, "ratio" is more appropriate.

Point 5: Sorry, with all explanations in the caption to Figure 2, I do not understand the meaning of the capital and small letters in this figure.

Response 5: Apologies for the confusion, I have corrected the manuscript and added more detail.